# Answer-state Recurrent Relational Network (AsRRN) for Constructed Response Assessment and Feedback Grouping

**Zhaohui Li**[1]     **Susan E. Lloyd**[2]     **Matthew D. Beckman**[2]     **Rebecca J. Passonneau**[1]

[1]Department of Computer Science and Engineering, Pennsylvania State University
{zjl5282,rjp49}@psu.edu
[2]Department of Statistics, Pennsylvania State University
{sel5591,mdb268}@psu.edu

## Abstract

STEM educators must trade off the ease of assessing selected response (SR) questions, like multiple choice, with constructed response (CR) questions, where students articulate their own reasoning. Our work addresses a CR type new to NLP but common in college STEM, consisting of multiple questions per context. To relate the context, the questions, the reference responses, and students' answers, we developed an Answer-state Recurrent Relational Network (AsRRN). In recurrent time-steps, relation vectors are learned for specific dependencies in a computational graph, where the nodes encode the distinct types of text input. AsRRN incorporates contrastive loss for better representation learning, which improves performance and supports student feedback. AsRRN was developed on a new dataset of 6,532 student responses to three, two-part CR questions. AsRRN outperforms classifiers based on LLMs, a previous relational network for CR questions, another graph neural network baseline, and few-shot learning with GPT-3.5. Ablation studies show the distinct contributions of AsRRN's dependency structure, the number of time steps in the recurrence, and the contrastive loss.

## 1 Introduction

STEM educators must trade off the ease of assessing selected response (SR) questions, like multiple choice, with constructed response (CR) questions, where students articulate their own reasoning. Natural Language Processing (NLP) research has turned increasingly towards assessment of constructed response (CR) questions in recent years, but models remain relatively simple, in part because datasets are either relatively small for neural methods, or proprietary. In addition, public datasets are limited to standalone questions. Timely feedback is important in the context of formative assessment throughout a course. As a result, formative assessments in STEM currently rely largely on SR items, where assessment is immediate. This conflicts, however, with the current emphasis in STEM on critical thinking, reasoning and writing (Graham et al., 2020; Birenbaum and Tatsuoka, 1987). Manual assessment of CR is time consuming, and more so when feedback comments accompany a score. We present a relational neural network that can be adapted to handle different CR formats, and that outperforms strong baselines on a new dataset of statistics questions developed to measure statistical reasoning at the introductory college level. Moreover, it also groups responses for similar feedback, in an unsupervised way.

The most well known and reasonably sized NLP datasets for CR questions are the BEETLE and SciEntsBank data from SemEval 13 Task 7 (Dzikovska et al., 2013) and the Kaggle dataset ASAP (Shermis, 2015). The former has 253 prompts across 16 STEM domains; the latter has 10 prompts where the STEM is mostly from biology, with 22,000 items. Assessment scales range from 2-way to 5-way. There are often multiple reference answers for each point level. ASAP also contains scoring rubrics. In both datasets, the prompts are standalone questions, but there are many other CR formats (Livingston, 2009; Bennett, 1991). We present ISTUDIO, a new dataset with context-based questions, a common format in STEM. Table 1 shows a context, its two questions, the rubric and reference answers for the first question, and a student response to each question.

Recent automatic assessment of CR questions using neural models has achieved good performance on the benchmark datasets mentioned above. The standard approach is to encode the question prompt and student answer separately, and send their concatenation to a classifier output layer. Riordan et al. (2017) achieved good performance on ASAP with a biLSTM. Later work used transformers (e.g., BERT (Devlin et al., 2019)). Using a model that encodes rubric elements, Wang et al. (2019) achieved

| Context 2 | Some people who have a good ear for music can identify the notes they hear when music is played. One method of note identification consists of a music teacher choosing one of seven notes (A, B, C, D, E, F, G) at random and playing it on the piano. The student is asked to name which note was played while standing in the room facing away from the piano so that he or she cannot see which note the teacher plays on the piano. |
|---|---|
| Questions | **[2a]** Should statistical inference be used to determine whether Carla has a "good ear for music"? Explain why you should or should not use statistical inference in this scenario. 
 **[2b]** Next, explain how you would decide whether Carla has a good ear for music using this method of note identification. |
| Rubric for [2a] | Student advocates use of statistical inference **AND** Student provides rationale with accommodation for variability (e.g. repeat test method many times; compare to chance model) **OR** Student describes analysis of probability/proportion/number of correct-incorrect. |
| Reference answers for [2a] | **[Correct (2 points)]** We should use statistical inference because there could be cases of Carla just getting lucky if we just count how many times she gets a note right. 
 **[Partially Correct (1 point)]** I would use statistical inference to see if Carla has a good ear for music. 
 **[Incorrect (0 points)]** No, there are no numerical values which could be applied to a "Good ear for music". |
| Student answers | **[Student_2CKz_q2a]** You could use statistical inference by seeing how many notes she can answer correctly and the more she gets then the better "ear for music" she has. **(Correct: 2 points)** 
 **[Student_2CKz_q2b]** You could give her a test on the notes and the more she answers correctly, then the better she is at music. **(Partially Correct: 1 point)** |

Table 1: An ISTUDIO example with a context and two question prompts. For [2a], the rubric for correct is shown (other classes omitted to save space), plus one reference example per three correctness classes. A student response is shown for each question.

results similar to (Riordan et al., 2017). Saha et al. (2018) separately encoded the question, the reference answer and the student answer using InferSent (Conneau et al., 2017), and combined the three vectors with manually engineered features for word overlap and part-of-speech information as input to a simple classifier. On the nine SemEval tasks, accuracies ranged from 0.51 to 0.79. Good results are also reported on proprietary datasets (Saha et al., 2018; Sung et al., 2019; Liu et al., 2019). Camus and Filighera (2020) compared several pre-trained models on SciEntsBank, where RoBERTa (Zhuang et al., 2021) performed best. SFRN (Li et al., 2021) outperformed all these models using a more novel relational architecture. It used BERT to separately encode the question, the reference answer (SemEval) or a rubric (ASAP), and the student answer, and then learned a single relational vector over the three encodings. We use a more complex relational network that outperforms SFRN.

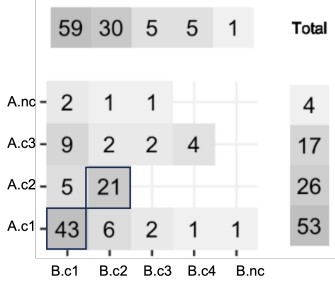

Figure 1: Coincidence matrix of feedback clusters identified by two statistics experts, A (3 clusters) and B (4 clusters) for 100 student responses to ISTUDIO question 2b (shown in Table 1; "nc" is for no cluster). The boxed cells show overlap in two clusters, amounting to 64% of the total.

Besides automating the assignment of a correctness score, NLP has the potential to support feedback comments. However, when and how to provide such feedback is not well understood, and is more of an unsupervised clustering problem than a classification problem, as illustrated in Figure 1. The figure shows a coincidence matrix of feedback clusters for 100 randomly selected student responses to ISTUDIO question [2b] labeled partially correct, and where answers were partially or essentially correct on [2a]. Two statistics experts worked independently to cluster the responses, and assign a feedback phrase. Expert A identified three clusters that covered all but 4 items (**nc** for no cluster). Expert B originally identified 14 that covered all but one item, then combined clusters that were similar and arrived at 4 clusters. Figure 1 shows an overlap of 43 items from A's and B's largest clusters, and another overlap of 21 items. All the other cells have small counts. This shows that different people are likely to provide comments in different ways, but might agree on a few categories that would cover a large proportion of student answers. To make progress towards automated formative assessment that can group similar responses, we develop a novel use of contrastive learning (Chen et al., 2020a; Chopra et al., 2005), to learn representations that are more similar to a reference answer that is distinct from other reference answers.

Relational networks (Palm et al., 2018), where graph nodes are input elements and edges are relation vectors, can have complex graph configurations, and are more efficient than graph attention or graph convolution (Nastase et al., 2015; Liu and

Zhou, 2020). We propose AsRRN, a family of recurrent relational networks where recurrence is over time, to learn distinct relations among the different inputs in CR datasets. Depending on the constituency of a dataset, each distinct input text type (e.g., scenario context, question prompt, reference answer, student answer, rubric) is encoded by a pretrained language model, and becomes a node in a graph network where edges are dependency relations. We find that the highest-performing dependency structure is the one that captures the semantic dependencies of the domain. A global state node is a dependent of all other nodes. We apply contrastive learning (CL) where reference examples serve as positive and negative prototypes. AsRRN outperforms multiple strong baselines on correctness classification, and to some degree can replicate the feedback grouping illustrated in Figure 1.

## 2 Related Work

Deep neural networks (DNNs) have been applied to CR assessment at least since (Alikaniotis et al., 2016). As noted in the introduction, most DNNs for CR encode the (reference answer, student answer) pair and use various output functions. Alikaniotis et al. (2016) used Long-Short Term Memory (LSTM) networks enhanced by word-specific scores, and achieved high score correlations on ASAP. Riordan et al. (2017) combined Convolutional Neural Networks (CNNs) with LSTMs. Model performance with these earlier DNNs was sometimes shown to improve in combination with hand-crafted features (Saha et al., 2018) and text mining (Süzen et al., 2020). As noted above, pretrained LMs like RoBERTa have performed well (Camus and Filighera, 2020). We rely on pretrained LMs to encode the input layer, and use a novel relational network.

Explicit use of rubric information is less common. AutoP achieves a performance boost on ASAP through automatic generation of regular expression patterns from top-tier answers and rubrics (Ramachandran and Foltz, 2015). Wang et al. (2019) integrate rubric information via word-level attention, finding that the rubric information compensated for less data, but did not otherwise boost performance. We found that incorporating rubrics into AsRRN did not improve performance on IS-TUDIO. This may be due to the compositional logic of rubric statements, that can have multiple conjunctions and disjunctions (see Table 1).

Relational networks (RNs) were initially developed as an alternative to other graph-based models for reasoning problems in visual, linguistic, or symbolic domains (e.g., physics (Santoro et al., 2017)), where distinct types of input elements have correspondingly distinct interrelations. RNs have surpassed human performance on the CLEVR visual question answering dataset (Johnson et al., 2017), and have become a general framework for few-shot learning (Sung et al., 2018). Li et al. (2021) showed that an RN for short answer assessment could efficiently learn relations among encodings of the question, reference answer, and student answer. Recurrent Relational Networks (RRNs) (Palm et al., 2018) eschew the flat relational structure of an RN for a fully connected relational graph with pairwise message passing. We found this original RRN structure to be disadvantageous for our CR tasks (see discussion accompanying Figure 3). For ISTUDIO, AsRRN adopts a dependency structure that reflects how people reason about the text elements shown in Fig. 1.

Contrastive Learning (CL) aims to improve representation learning by making similar data instances closer and separating dissimilar ones (Le-Khac et al., 2020). CL can be applied in unsupervised or supervised learning, and benefits a wide array of visual, language and multi-modal tasks. Chopra et al. (2005) pioneered the introduction of contrastive loss in dimensionality reduction. Subsequently, Schroff et al. (2015) proposed triplet loss, which employs an anchor to concurrently handle positive and negative samples. Later, Chen et al. (2020b) introduced the highly-regarded self-supervised normalized temperature-scaled cross entropy loss (NT-Xent), with strong results on unsupervised learning in ImageNet. Khosla et al. (2020) extended NT-Xent loss for application to supervised learning; CERT (Fang et al., 2020) improved pretrained language models (e.g., BERT) using contrastive self-supervised learning at the sentence level. Qiu et al. (2021) and Pan et al. (2022) developed new text classification methods using contrastive learning loss, and SimCSE (Gao et al., 2021) delivered state-of-the-art sentence embeddings on semantic textual similarity (STS) tasks. We propose a variant of NT-Xent with a distinct treatment for different classes: we assume multiple correct reference examples to the same question should be **similar**, whereas different ways to be partially correct should be **dissimilar** (see Figure 1).

## 3 Dataset

The context-based format is a novel challenge for NLP, but is widely used in STEM, e.g., in so-called two-tier assessments that simultaneously address students' content knowledge/skills in tier one and explanatory reasoning in tier two, as in computer science (Yang et al., 2015), mathematics (Hilton et al., 2013), physics (Xiao et al., 2018) and other STEM subjects. Automatic methods for low-stakes formative assessment can support timely feedback to identify which conceptual issues to address next, thus can lead to greater student success in STEM. CR questions that address reasoning skills are preferable to selected response items but burdensome for humans to assess. The ISTUDIO data come from an assessment instrument for statistical reasoning called *The Introductory Statistics Transfer of Understanding and Discernment Outcomes (ISTUDIO)* that was found to be educationally valid and reliable (Beckman, 2015), meaning there is strong evidence that it measures well what it intends to measure.

The ISTUDIO dataset consists of responses from 1,935 students to three 2-part questions. The assessment rubric, which uses a 3-way scale of *(essentially) correct, partially correct, incorrect*, specifies the scoring criteria and provides example student answers for each correctness class. The data is de-identified, and the original IRB confirmed that it can be made public for research purposes in its de-identified form. Students' schools of origin are identified as large, medium or small.[1] We evaluated inter-rater agreement among two of the co-authors (A, B), and a graduate student in statistics (C). There were 63 items scored by A, B, and C; two additional sets of 63 were scored by A, B, and B, C. These pairs of raters achieved quadratic weighted kappa (QWK) > 0.79, and Fleiss Kappa showed substantial agreement among A, B and C (FK = 0.70). The remaining unlabeled data was assessed independently by A, B and C.[2]

To prepare ISTUDIO for NLP research, we performed data cleaning to eliminate non-answers and responses that were unusually long (greater than 125 word tokens). This yielded 7,258 individual student response items, ten percent of which was held in reserve for future research. Ninety percent of the remainder (N=6,532) was partitioned in the proportion 8:1:1 into training (N=5,226), validation (N=653), and test (N=653) sets. The test set contains all the responses with labels from multiple raters so that at test time, inter-rater agreement of model predictions can be computed with the reliable human labels.

## 4 Approach

The ISTUDIO dataset contains four distinct types of textual input: the context $C$ for a given question prompt, the question prompt $Q$, one or more reference answers $R$ for each correctness class $c \in \{0, 1, 2\}$, and the student answers $A$.[3] Among many possible ways to structure the set of encodings of each text type–such as concatenation, stacking, attention layers across sequences of text types, and graph networks–we hypothesized that semantic relations among the five text types could be structured as a dependency graph rooted at $C$, with the learned representation of each dependent node being directly influenced by its parent, and indirectly by its ancestors. As discussed in detail below, the specific chain of dependencies we hypothesized is shown in Figure 2. Our hypothesis is similar in spirit to the motivation for recurrent relational networks, namely to process a *chain of interdependent steps of relational inference*, as in (Palm et al., 2018; Battaglia et al., 2018). Here, recurrence refers to inferential time-steps for the entire graph, rather than recurrence of processing units. Temporal recurrence allows the network to iteratively progress towards the full set of relations among all the nodes of the graph (Palm et al., 2018). AsRRN also incorporates a super-node that is dependent on all other nodes, as in (Zhang et al., 2018), to stabilize the flow of information in the graph network, and to implicitly represent the current answer state at each pass over the whole graph. We test our hypothesized relational structure among $C, Q, (R_{rubic}, ), R$ and $A$ through extensive ablations of alternative dependency structures, and by comparison to strong baselines that either have a simpler relational structure, or no relational structure at all. In this section, we present details on the graph structure, the flow of information within the graph (message-passing), the temporal recurrence, the output classifier layer, and our training

---

[1]Li, Z., S. E. Lloyd, M. D. Beckman, and R. J. Passonneau. ISTUDIO: A Statistical Automatic Response Assessment Dataset. Penn State Data Commons, 2023. https://doi.org/10.26208/JFMP-V777

[2]Fleiss Kappa is described in (Fleiss, 1971). QWK applies a quadratic weighting for agreement, partial agreement, and disagreement applied to Cohen's Kappa (Cohen, 1960).

[3]Incorporating the rubric text into AsRRN was not helpful.

objective that incorporates contrastive learning.

**AsRRN's relational graph** for ISTUDIO has the four layers shown in schematic form in Figure 2. The input layer to each node type relies on a pre-trained model to encode the distinct types of text input at time $t_0$. Note that the $C$, $Q$, and $A$ layers are shown as consisting of a single node, whereas the $R$ layer has four nodes. This indicates that the pretrained encoding of a single text input is used to initialize the layers for $C$, $Q$ and $A$ whereas for the $R$ layer we initialize with multiple, different reference answers for each correctness class. Again through experimentation, we found that four reference answers for each correctness class achieved the best performance. For simplicity, the figure shows only four $R$ nodes instead of the full set of twelve per $Q$ (four per class). The arrows showing the dependencies between layers (referred to below as message passing) essentially correspond to relation vectors computed from the hidden states at each node. The super-node $S$ is a global relational vector that incorporates information from the nodes that correspond to the distinct text inputs. The figure also illustrates the temporal recurrence for updating the flow of information within the graph.

Before describing the information flow, or message passing, we point out here that AsRRN can be adapted for CR datasets that have different structures. For example, the $C$ layer can be removed to fit the types of standalone question prompts found in benchmark datasets such as SemEval-2013 Task 7 (Dzikovska et al., 2013) and ASAP (Shermis, 2015). This is how we perform AsRRN experiments on these two benchmarks, as reported below.

**AsRRN's Recurrent message passing over time** corresponds to the computation of relational information among nodes within the neural network graph structure. In AsRRN, message passing is restricted to pairs of nodes $i$ and $j$ at time $t$, and each *message* $m_{ij}^t$ (or relational vector) is:

$$m_{ij}^t = f\left(h_i^{t-1}, h_j^{t-1}\right) \qquad (1)$$

where $f$, the message function, is a multi-layer perceptron. All messages $m_{ij}^t$ are dispatched in parallel, including the super-node. To update each node $i$, the messages from its set of neighbor nodes $N(i)$ are first summed: $m_i^t = \sum_{j \in N(i)} m_{ij}^t$. Each node hidden state $h_i^t$ is updated recurrently:

$$h_i^t = g\left(h_i^{t-1}, x_i, m_i^t\right) \qquad (2)$$

where the node function $g$ is another MLP, and $x_i$ is the original input encoding at node $i$. Reliance on each node's previous hidden state $h_i^{t-1}$ allows the network to iteratively work towards a solution, as described in (Palm et al., 2018). Further, persistent use of the input feature vector $x_i$ at each time step is similar to the operation of a residual network (He et al., 2016), relieving $g$ from the need to remember the input, and allowing it to focus primarily on the incoming messages from $N(i)$.

**Classification** of a student's answer for correctness utilizes the hidden state at node $A$ at the final time step $t$. The output distribution $o^t$ is:

$$o^t = \Theta\left(A^t\right) \qquad (3)$$

where $\Theta$ is a learnable function to assign the $A$ node's hidden state to class probabilities.

The **training objective** includes cross entropy loss for classification performance and a new contrastive learning loss that exploits the availability of reference answers to improve the representation learning. Contrastive learning aims to reduce the distance between learned representations of positive examples to a set of positive exemplars, and to increase the distance from a set of negative exemplars (Chopra et al., 2005). We adapt the widely used NT-Xent contrastive loss function (normalized temperature scaled cross entropy; see above).

Here, where we have $n$ reference examples for each of the three correctness classes, we treat the $n$ reference answers for the predicted class as positive exemplars, and the $2n$ reference answers for the other two classes as negative exemplars.

Our CL function depends on the true class label. For our use case, we make three strong assumptions: 1) that there is **one way** for a student answer to be correct regardless of the exact wording, 2) that there are **a few ways, each distinct from the others**, for a student's answer to be partially correct that will cover many of the answers, and 3) that there are many diverse ways to be incorrect. For the correct class, the $n$ reference answers are assumed to be near paraphrases, thus for a student answer $x_i$ whose label is *correct*, $L_{CL}$ incorporates a term $S_j$ for the **average** of its similarities to the $n$ correct reference examples:

$$\mathcal{S}_j = \text{Average}\left[\frac{sim(\boldsymbol{x}_i, \boldsymbol{x}_{j_1})}{\tau}, \ldots, \frac{sim(\boldsymbol{x}_i, \boldsymbol{x}_{j_n})}{\tau}\right]$$

where $x_j$ are exemplars from the same class as $x_i$, and $x_k$ are exemplars from the other classes.

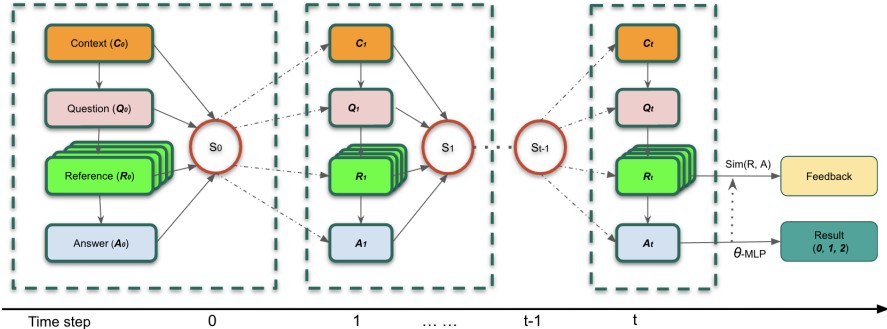

Figure 2: For ISTUDIO, the AsRRN dependencies are C→ Q, Q→R, R→A, as shown in the figure. During training, AsRRN uses message passing along the arrows to update each node, includng a global supernode. At the final time step, a learnable function $\Theta$ computes the output probability distribution. The maximum cosine similarity of vectors $A$ to the four reference answers $R$ in the predicted class supports potential feedback.

For items $x_i$ predicted to be partially correct or incorrect, our loss function incorporates a term $S'_j$ for the **maximum** of its similarities to the $n$ reference examples in the same class:

$$\mathcal{S}'_j = \text{Max}\left[ \frac{sim(\boldsymbol{x}_i, \boldsymbol{x}_{j_1})}{\tau}, \ldots, \frac{sim(\boldsymbol{x}_i, \boldsymbol{x}_{j_n})}{\tau} \right]$$

This ensures that if there is one partially correct reference answer that the student answer resembles, the learned representation will become closer to that reference answer.

All class predictions use a normalizing term $S_k$:

$$\mathcal{S}_k = \sum_k \frac{sim(\boldsymbol{x}_i, \boldsymbol{x}_k)}{\tau}$$

yielding two contrastive loss terms, depending on the ground truth class label:

$$\mathcal{L}_{CL} = -\frac{1}{N} \sum log \frac{exp(\mathcal{S}_j)}{exp(\mathcal{S}_j) + exp(\mathcal{S}_k)} \quad (4)$$

$$\mathcal{L}'_{CL} = -\frac{1}{N} \sum log \frac{exp(\mathcal{S}'_j)}{exp(\mathcal{S}'_j) + exp(\mathcal{S}_k)} \quad (5)$$

In sum, correct predictions are encouraged to be similar to the average of the correct reference examples and dissimilar to the reference examples for partially correct and incorrect. Partially correct (and incorrect) predictions are encouraged to be similar to one of the partially correct (or incorrect) predictions, and dissimilar to the reference examples for the other two classes.

The full loss function is a weighted sum of the cross entropy loss and the contrastive loss: $L = (1 - \lambda)L_{CE} + \lambda L_{CL}$, where $\lambda$ is a hyperparameter that governs the proportion of the two types of loss. Cross entropy loss is:

$$L_{CE} = -\frac{1}{N} \sum_{i=1}^{N} \sum_{j=1}^{C} y_{ij} \log(p_{ij}) \quad (6)$$

where $y_{ij}$ is the true label for the $i$th sample and $j$th class, and $p_{ij}$ represents the predicted probability for the $i$th sample and $j$th class. $N$ is the number of samples, and $C$ is the number of classes. If sample $i$ belongs to class $j$, $y_{ij}$ is 1 (true label) and 0 otherwise, and $p_{ij}$ is the predicted probability that sample $i$ belongs to class $j$.

## 5 Experiments

Four experiments are reported here. In the first, AsRRN significantly outperforms multiple baselines on ISTUDIO, the 3-way unseen answers (UA) subset of BEETLE and SciEntsBank, and the 3-way subset of ASAP. The second set compares AsRRN to chain-of-thought prompting on GPT-3.5. Third, extensive ablations verify the benefits of multiple design choices in AsRRN. Fourth, we compare As-RRN's grouping of student answers with human grouping for partially correct responses.

For baselines, we use four high-performing LMs available on Huggingface (BERT-base-uncased; RoBERTa-base; ConvBERT-base; DistilBERT-base-uncased); also SFRN (Li et al., 2021) and a benchmark graph neural network model graph attention network (GAT) (Veličković et al., 2017). For the LMs, we concatenate the encodings of the question prompt, reference answer, and student answer, and add an MLP output layer followed by softmax. We do not encode the context in part because it sometimes exceeds the input token length and in part because it does not seem to improve performance. We use two versions of SFRN, with BERT encodings at the input layer; we either pass the 4-tuple of the context (C), question (Q), reference answer (R) and student answer (A) to the relational layer, or the (Q,R,A) triple (as in (Li et al., 2021). For the GAT experiments, we also

| Model | ISTUDIO | BEETLE | SciEntsBank | ASAP |
|---|---|---|---|---|
| BERT-base-uncased | 80.44 (78.71, 81.78) | 75.21 (73.91, 77.17) | 70.74 (69.19, 71.70) | 77.43 (76.33, 78.10) |
| roBERTa-base | 81.30 (77.34, 83.15) | 75.78 (73.60, 78.73) | 69.43 (66.96, 70.96) | 76.28 (75.42, 77.35) |
| ConvBERT-base | 80.49 (78.87, 82.54) | 75.41 (73.91, 77.33) | 69.83 (66.52, 71.74) | 78.64 (78.01, 79.01) |
| distilBERT-base-uncased | 79.71 (78.71, 81.47) | 75.21 (73.14, 78.57) | 69.94 (67.85, 72.00) | 76.82 (75.42, 77.35) |
| GAT (4 heads) | 78.48 (76.72, 80.25) | 74.30 (72.52, 75.62) | 69.31 (68.15, 70.22) | 77.17 (76.15, 79.16) |
| GAT (8 heads) | 78.91 (75.34, 81.16) | 74.81 (71.89, 76.86) | 67.31 (65.44, 68.74) | 78.41 (77.21, 79.47) |
| SFRN (C,Q,R,A) | 81.68 (79.26, 83.47) | N.A. | N.A. | 78.94 (78.02, 80.19) |
| SFRN (Q,R,A) | 82.15 (80.96, 83.83) | **77.02 (75.47, 78.88)** | **71.74 (68.30, 73.59)** | 78.75 (78.02, 79.40) |
| AsRRN (-CL) | 83.46 (82.15, 84.23) | 76.71 (75.47, 77.64) | **72.11 (71.41, 73.04)** | **80.02 (79.14, 81.20)** |
| AsRRN (+CL) | **85.26 (83.46, 86.98)** | N.A. | N.A. | N.A. |

Table 2: AsRRN and baseline accuracies (CIs) on three test sets, and QWK on the fourth. SFRN (C,Q,R,A) cells marked N.A. indicate the dataset has no context. AsRRN (+CL) cells marked N.A. are for datasets with only *correct* reference answers.

| Model | Accuracy | | | | | | |
|---|---|---|---|---|---|---|---|
| | 2.a | 2.b | 3.a | 3.b | 4.a | 4.b | Total |
| GPT-3.5: -Ref, -Rub | 47.17 | 57.00 | 29.35 | 40.59 | 57.01 | 74.54 | 51.12 |
| GPT-3.5: +RefC, +RubC | 49.10 | 32.71 | 67.88 | 58.41 | 50.00 | 75.45 | 55.59 |
| GPT-3.5: +2RefAll, +RubAll | 56.25 | 38.31 | 65.13 | 64.35 | 70.17 | 81.81 | 62.78 |
| AsRRN | **83.92** | **84.11** | **92.66** | **83.16** | **88.59** | **89.09** | **85.26** |

Table 3: AsRRN model best performance compared with GPT3.5 prompting.

use BERT pretrained embedding as the input of each node, the graph structure we use is the default connections of all C, Q, R, and A nodes with 4 or 8 self-attention heads.[4] All results show 95% bootstrap confidence intervals (Efron and Tibshirani, 1986) from 8 runs on the test sets (parallelized over 8 GPU cores). Table 2 shows evidence of stastical differences, with AsRRN outperforming the four LMs. AsRRN (+CL) also outperforms SFRN (C,Q,R,A), and nearly so for SFRN (Q, R, A). The AsRRN model not only outperforms the GAT model across all datasets, but it also boasts a shorter running time. This underscores AsRRN's superior efficiency and enhanced reasoning capacity. Details on the AsRRN settings are below; training parameters are in appendix A.

GPT models often perform well using chain-of-thought reasoning for zero-shot and few-shot settings(Brown et al., 2020; Wei et al., 2022), so we tested this method on ISTUDIO, using GPT-3.5. In all conditions in Table 3, showing total accuracy and accuracies per question, the prompting included the context, the question, the class labels, and a student answer for the entire test set (Appendix D includes details). Results in the first row (-Ref, -Rub) are from these inputs alone (similar to zero-shot). In the second row, one correct reference answer and the rubric for the correct class were added (+1RefC, +RubC). In the third row condition, two reference examples for each class, and the rubric for each class were provided (+2Re-

fAll, +RubAll). GPT-3.5 improves in each next condition, and for question 4.b, comes close to the LM performance. Interestingly, the relative performance by question for GPT-3.5 is quite different from AsRRN, and both are quite different from the relative difficulty for students.

Table 2 above shows that ablation of CL degrades performance. We conducted multiple ablations on ISTUDIO, and investigated the impact of replacing CL with a regularization term. We tested the contribution of each AsRRN layer, finding that all layers contribute significantly to the overall performance (results not shown due to space constraints). We also tested different numbers of reference answers and found that four worked best. The four reference answers for each of the 6 questions were chosen by the four co-authors (2 NLP researchers, 2 researchers in statistics education) to ensure they were distinct from each other, and covered many training examples. Figure 3 shows the mean accuracy with different values for the temporal recurrence, and different graph structures. The blue line (circles) for AsRRN shows that learning continues to improve over the first two time steps, then degrades. The red line (triangles) for a fully connected RRN is worse at all but the first time step. The yellow line (boxes), labeled random, represents that in each of the 8 parallelized CI runs, for all pairs of nodes $i, j$ (cf. equation 1), a random decision was made at the beginning of training whether to include message passing; a full exploration of all combinations of message passing

---
[4]Less fully connected configurations performed worse.

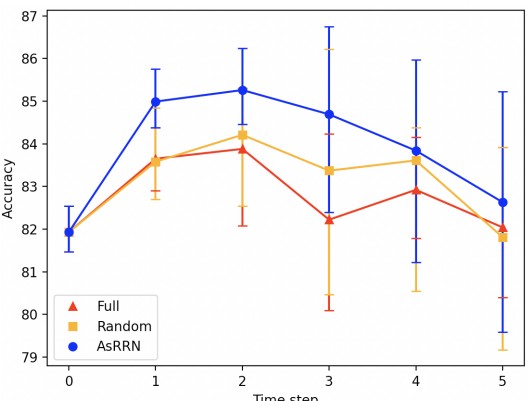

Figure 3: Mean test accuracy and CIs with different numbers of reasoning time steps for AsRRN, a fully connected RRN, and an RRN with random message passing.

would have been unnecessarily time-consuming. This condition has lower performance than AsRRN across all but the last time step. Finally, we examined different values of $\lambda$ in the loss function, and compared to L1 or L2 regularization. For all values of $\lambda < 1$, performance is better than for $\lambda = 1$ (see Appendix C). For $\lambda = 1$, accuracy is still reasonable, because $L_{CL}$ implicitly includes the ground truth class. Replacing $L_{CL}$ with an L1 or L2 regularizer severely degrades performance. Results in Tables 2-3 use 4 reference answers per class, 2 time steps, and $\lambda$=0.01.

Evidence suggests that timely and specific feedback helps students learn (Pearl et al., 2012; Garfield et al., 2008). For formative assessment feedback, instructor effort is best spent on partially correct (PC) answers, where there is an opportunity to scaffold students towards more complete understanding. Incorrect answers are far more diverse, and students may be more confused about the reasoning steps and less likely to integrate feedback into their thinking. To assess the results on whether AsRRN groups student responses with the same reference answers human would, we find mixed results when we compare two humans and AsRRN on the grouping of partially correct answers. Recall that there were four reference answers for all three correctness classes, for all six questions. For the

partially correct class, we selected four reference answers that were distinct from one another. Here we report on how consistent two experts (A and B) were with each other, and each with AsRRN, in pairing a student answer with the same reference answer for questions 2b and 4b; these are two of the three most challenging questions, based on the average student scores. We used QWK to report agreement of the three pairs. For both questions, A and B worked independently to pick one of the four reference answers as equivalent to each students' *partially correct* answer, or "none of the above." For the subset of accurate partially correct predictions from AsRRN (90 out of 107 for 2b; 100 out of 110 for 4b), we show QWK for the expert pair, and each expert with AsRRN. Cases where AsRRN's maximum probability class had $p < 0.85$ were treated as "none of the above" based on locations of sharp decreases of output probabilities in scatterplots.

As shown by the QWK scores in Table 4, the two experts had good agreement of 0.72 on question 2b, which is consistent with the pattern shown in the coincidence matrix in Figure 1. They had very poor agreement on 4b, however, which shows that this task is difficult even for experts. On 2b, AsRRN agreement with B of 0.69 was close to that for A and B on the partially correct reponses that AsRRN accurately predicted (0.71), but agreement with A was much lower (0.53). On 4b, agreement between A and B on AsRRN's accurate subset was 0.45, and here AsRRN had similar agreement with B of 0.41, then very high agreement with A of 0.70. Instructors report that it is unpredictable with any given set of students what kinds of partially correct answers will occur frequently, as it depends on many unknowns, such as the students' prior background. Using contrastive learning in a novel way, we have taken a first step towards partially automating this difficult but pedagogically important task.

The high accuracy of AsRRN, and its potentially useful association of student answers with a specific reference answer, argues for the quality of the representation learning. Appendix B includes t-SNE visualizations of the learned representations of the test set with and without contrastive learning.

## 6   Conclusion

AsRRN is a novel recurrent relational network where the nodes in the neural graph each correspond to a distinct type of textual input, a super-

| Q | Test Set | | Correct Pred. | | | |
|---|---|---|---|---|---|---|
| | A, B | Size | A, B | A, RN | B, RN | Size |
| 2b | 0.72 | 107 | 0.71 | 0.53 | 0.69 | 90 |
| 4b | 0.39 | 110 | 0.45 | 0.70 | 0.41 | 100 |

Table 4: For questions 2b and 4b, QWK for experts on the test set; also QWKs on the subset of AsRRN's correct predictions for the experts and AsRRN (RN in the table).

node represents global information across all nodes, and the answer state at the final time step becomes the input to the final classification layer. The graph structure (direction of message passing), leads to efficient learning of relational information that supports multi-step problems. AsRRN can have more or fewer layers, depending on the number of distinct types of text input in the dataset. A novel loss function for contrastive learning improves classification accuracy, and provides a step towards feedback for instructors and students on partially correct responses, by learning which of multiple reference answers a response is most similar to. A thorough set of ablations provides evidence for the chosen dependency structure, number of time steps in the temporal recurrence, and the contrastive learning. On ISTUDIO, AsRRN significantly outperforms strong baselines.

## 7 Limitations

The AsRRN model relies heavily on manual graph design, based on the relational structure of the CR dataset, and on the quality of reference answers. While this limits generalizability, it also incorporates expert domain knowledge in useful ways. Additionally, while AsRRN provides a preliminary foundation for feedback, understanding of optimal feedback across varying assessment contexts remains limited. Although the importance of timely and relevant feedback for students is well-established, little is known about the specific forms of feedback that work best in different contexts, or how feedback varies across instructors. While AsRRN can point to a specific reference answer for partially correct student answers, the task definition and AsRRN performance need much improvement.

The ISTUDIO dataset has relatively few question prompts (N=6) and is limited to a single subject domain, in contrast to benchmark datasets that have dozens of prompts and many domains. On the other hand, ISTUDIO is derived from a well-validated, reliable instrument and was labeled through an extensive inter-rater agreement study.

## 8 Acknowledgements

This work was supported under NSF DRK award 2010351 and NSF IUSE award 2236150.

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

## A   Training Parameters

We employed the AdamW optimizer for model training, with parameters set to *learning_rate=1e-5* and *weight_decay=0.01*, and clipped the global norm of the gradient at 1.0. The state vector of the graph nodes and the message passing vector have hidden dimensions of 768 and 128, respectively. The weight of the CL loss ($\lambda$) is set to 0.01, and the temperature ($\tau$) of CL loss is 1. To curb overfitting, a 20% dropout rate (*hidden_dropout_prob=0.2*) is applied to the hidden layers. The maximum token length for each text data input is set to 64 in BEETLE dataset, 128 in SciEntsBank and ASAP datasets, and 256 in ISTUDIO.

We use warm-up prior to training with no CL loss during which the learning rate incrementally increases from a negligible value to the pre-set learning rate. The *get_linear_schedule_with_warmup*

function from the Hugging Face Transformers library generates a schedule with a learning rate that linearly decreases following a warmup period of linear increase. In our experiment, the number of warmup steps is configured to be 5% of the total training steps.

All models were trained on a server equipped with eight NVIDIA RTX A5000 GPU cores, with the AsRRN model's training time ranging from two to five hours, depending on the training set size and number of epochs. If the paper is accepted, we will include a github link to the code repository.

## B Contrastive Learning Visualization

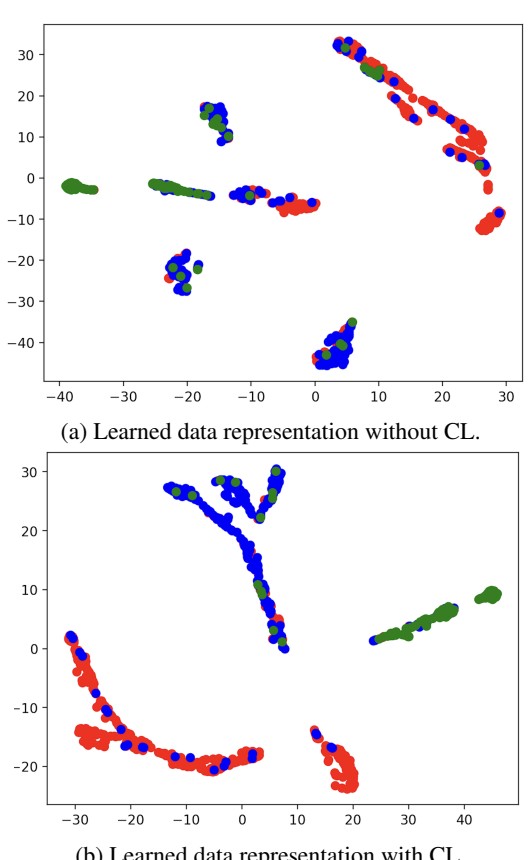

(a) Learned data representation without CL.

(b) Learned data representation with CL.

Figure 4: Learned representations without (a) and with (b) contrastive loss, visualized with t-SNE projections into 2D. The color key is green for correct, blue for partially correct, and red for incorrect.

T-SNE projections of the representations of the examples in the test set are shown in Figure 4 for the two conditions of omitting the contrastive loss term (4a) versus including it (4a). With the contrastive learning, the items cluster more tightly into fewer groups.

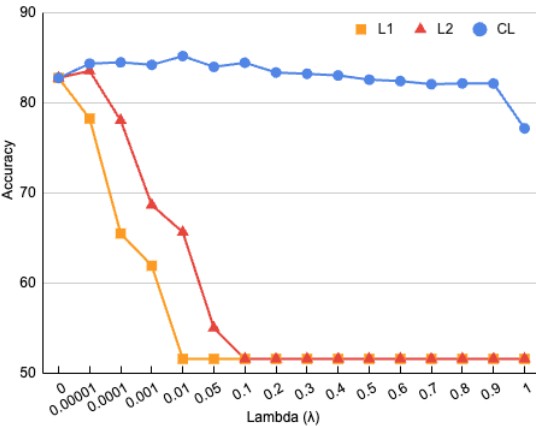

Figure 5: Mean test accuracy of AsRRN on ISTUDIO with different $\lambda$ values in the loss function. CL represents the contrastive learning in the loss function. Using L1 or L2 regularization instead of CL in the combined function degrades performance.

## C Contrastive Loss versus Regularization

For values of $\lambda < 1$, Figure 5 shows that the overall AsRRN accuracy is fairly stable. For $\lambda - 1$ accuracy drops to around 78%, which is still reasonable. This can be attributed to the diminished supervision of the classification loss, and the the fact that the contrastive learning loss employs ground truth labels during training, using the **mean** or **max** similarity to reference examples in the ground truth class.

## D GPT 3.5 Prompting

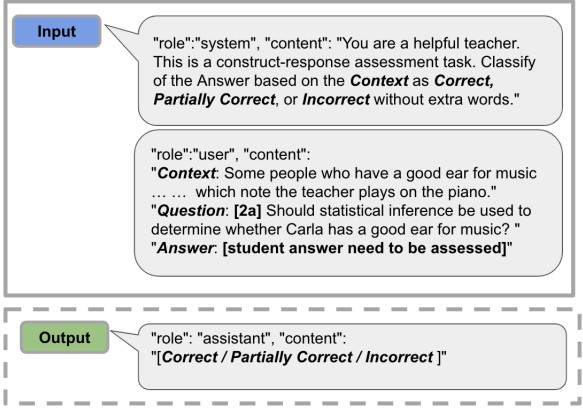

Figure 6: Zero-shot learning prompt design for the assessment of ISTUDIO data. Input is the prompt message, Output is the model responses

GPT 3.5 is a powerful model for generating human-like text. In our experiment, we leveraged GPT 3.5 model with a series of "prompts" based on the ISTUDIO materials (containing Context (C), Question (Q), Reference answer (R), and Answer

(A)). We articulated these prompts using a chain-of-thought strategy. Our GPT-3.5 model was implemented using the *ChatCompletion* API provided by OpenAI, specifically using the *gpt-3.5-turbo* model with a 0.7 model temperature.

As exemplified by [2a] in Figure 6, we designed a zero-shot learning prompt template for GPT-3.5 (-Ref, -Rub) displayed in Table 3. The model was supplied with a series of "messages" to generate classification results. Each message consisted of a role (either 'system', 'user', or 'assistant') and the message content associated with that role. In one or few-shot learning scenarios (Figure 7) where reference answers were added by the 'user', we provided historical examples to the input. For instance, the one-shot (+1RefC, +RubC) included a correct reference answer as an example in the prompt as shown in 7a. In the case of few-shot prompts (+2RefAll, +RubAll), we used six reference answers (two for each class) as depicted in 7b. However, due to space limitations, only three examples are shown. Finally, we ensured the model's output always included a correctness label followed by an explanation (all marked in red). This chain-of-thought prompting method not only enabled improved results but also laid the foundation for future research in feedback generation.

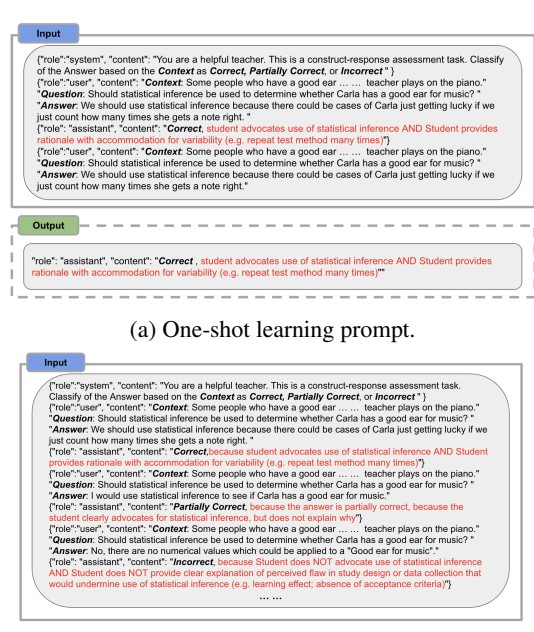

(a) One-shot learning prompt.

(b) Few-shot learning prompt.

Figure 7: One-shot and few-shot learning prompt examples for the assessment of ISTUDIO question [2a].