# OpenReview forum: "Answer-state Recurrent Relational Network (AsRRN) for Constructed Response Assessment and Feedback Grouping"
_EMNLP/2023/Conference — EMNLP 2023 Findings_

### Official Review · Reviewer_tuki · 2023-07-31

**Soundness:** 2

**Excitement:**

3: Ambivalent: It has merits (e.g., it reports state-of-the-art results, the idea is nice), but there are key weaknesses (e.g., it describes incremental work), and it can significantly benefit from another round of revision. However, I won't object to accepting it if my co-reviewers champion it.

**Paper Topic And Main Contributions:**

This work proposed an answer-state recurrent neural network to improve the quality of assessment of constructed response, which is a kind of scoring task for a given several inputs. In the proposed method, each input type (e.g., context, question, student answer, etc.) is encoded by a pre-trained language model, and the encoded results are fed into a variant of the recurrent network to make the states more suitable for the final prediction. In the training of the network, contrastive learning loss is introduced in addition to general cross entropy loss. To evaluate the proposed network in more STEM-specific setting, the authors created a new dataset Col-STAT, which consists of five input types, i.e., scenario context, question prompt, reference answer, student answer, and rubric. The evaluations were conducted using Col-STAT, BEETLE, SciEntsBank and ASAP datasets and the results showed that the proposed network with contrastive learning achieved the best results on Col-STAT and the proposed method without contrastive learning achieved slightly better performance than the baseline SFRN in three out of four datasets. The authors also compared their results with those obtained GPT-3.5.

**Questions For The Authors:**

- Question A: why did not the author show the results using rublic? To avoid the suspicion of arbitrarily changing the experimental setup, they should have first shown the results of the experiment using all input types and then the ablation results.

**Reasons To Accept:**

- The research motivation and the relationship between background and the proposed method are appropriately written in this paper.

- Although the performance is not much improved when contrastive learning is not used, the proposed graph-based approach may provide good inspiration for other researchers,

**Reasons To Reject:**

- The usefulness of the proposed recurrent neural network is unclear because the main factor of performance improvement is the introduction of contrastive learning (CL). Table 2 shows that the proposed method without CL slightly outperformed the baseline SFRN in three out of the four evaluation datasets, but some of them are not significant based on the 95% confidence intervals shown in Table 2. It would have been helpful to understand the usefulness of the combination of the graph-based recurrent neural network and contrastive learning in the proposed method if the results of contrastive learning in a simpler setting (e.g., contrastive learning is used when all input types are simply concatenated) had been included, but such results were not included in the paper.

- The difficulty in mapping the symbols (e.g., C_0, C_1, etc.) in Figure 2 to those in the equations makes it difficult to understand the details of the proposed method. For example, the authors did not explain what C_i, Q_i, R_i and A_i in Figure 2 mean in the main context, making it difficult to know how to compute each node in Figure 2. In addition, to my understanding, S_i in Figure 2 is different from the calligraphic S_j (line 431), but if so, how to compute S_i was not explained in this paper.

- The details of how the hyper-parameters of the proposed method were determined are unclear. For example, when predicting the final output using the proposed method, it is necessary to fix the time step t in advance, but how to determine t was not explained. If this t was determined using the validation data, such details should be noted in the paper to present that the experiments were properly conducted.

- It is difficult to understand STEM-related technical terms such as ISUDIO. Since most EMNLP readers are NLP experts but most of them are not experts of education research, technical terms specific to education research should be appropriately explained with examples. Particularly, the ITUDIO contexts should be explained with examples because no one can imagine the examples. In addition, I cannot understand what the difference is between "student response" (line 251) and "student response items" (line 289). To my understanding, the authors collected 1,935 "student responses", but after data cleaning, they obtained 7,258 "student response items". It is not clear why the data cleaning increased the data size.

**Reproducibility:**

3: Could reproduce the results with some difficulty. The settings of parameters are underspecified or subjectively determined; the training/evaluation data are not widely available.

**Reviewer Confidence:**

1: Not my area, or paper was hard for me to understand. My evaluation is just an educated guess.

**Typos Grammar Style And Presentation Improvements:**

- The full name of the proposed method (i.e., "Answer-state Recurrent Relational Network") only appeared in the Abstract. It also should be written in the main text of the paper.

- The font size in Figure 2 is too small to understand what the figure means. In addition, the positions of Figure 2 and its description are too far apart.

---

> ### Author Rebuttal · Authors · 2023-08-29
>
> Thanks for all your comments. We respond to your comments and questions below.
>
> “The usefulness of the proposed recurrent neural network is unclear because the main factor of performance improvement is the introduction of contrastive learning (CL). Table 2 shows that the proposed method without CL slightly outperformed the baseline SFRN in three out of the four evaluation datasets, but some of them are not significant based on the 95% confidence intervals shown in Table 2. “
>
>
> “It would have been helpful to understand the usefulness of the combination of the graph-based recurrent neural network and contrastive learning in the proposed method if the results of contrastive learning in a simpler setting (e.g., contrastive learning is used when all input types are simply concatenated) had been included, but such results were not included in the paper.”
>
> The results for AsRRN without contrastive learning (CL) and the two SFRN models are not statistically significantly different, as their confidence intervals overlap. The case remains that AsRRN (+CL) significantly outperforms all the LLM models, while none of the other relation networks significantly outperforms them. It is a great point that we should have applied CL to all the baselines. With SFRN, this would mean prior to the fusion operation, and we could communicate with the authors about revising the codebase that way. With the LLM baselines, it would be difficult to incorporate CL because we concatenated the text inputs.
>
>
> “The difficulty in mapping the symbols (e.g., C_0, C_1, etc.) in Figure 2 to those in the equations makes it difficult to understand the details of the proposed method. For example, the authors did not explain what C_i, Q_i, R_i and A_i in Figure 2 mean in the main context, making it difficult to know how to compute each node in Figure 2. In addition, to my understanding, S_i in Figure 2 is different from the calligraphic S_j (line 431), but if so, how to compute S_i was not explained in this paper.”
>
> Thank you for highlighting these issues. It appears we did not adequately explain the notation in Figure 2. In this figure, C_i, Q_i, R_i and A_i correspond to the components described in the Graph Building section of the paper the context (C), question (Q), reference answer (R), and the answer to be graded (A_i), respectively.
> The x_i in the equations serves as the node representation for A_i, while the other nodes C, Q, R are represented by h_j and x_j etc.
>
> “The details of how the hyper-parameters of the proposed method were determined are unclear. For example, when predicting the final output using the proposed method, it is necessary to fix the time step t in advance, but how to determine t was not explained. If this t was determined using the validation data, such details should be noted in the paper to present that the experiments were properly conducted.”
>
> Details about the hyper-parameters are in appendix A, and the time-step analysis is in Figure 3. Because time-step=2 had the best performance, we use t=2 at inference time.
>
> “It is difficult to understand STEM-related technical terms such as ISUDIO. Since most EMNLP readers are NLP experts but most of them are not experts of education research, technical terms specific to education research should be appropriately explained with examples. Particularly, the ITUDIO contexts should be explained with examples because no one can imagine the examples. In addition, I cannot understand what the difference is between "student response" (line 251) and "student response items" (line 289). To my understanding, the authors collected 1,935 "student responses", but after data cleaning, they obtained 7,258 "student response items". It is not clear why the data cleaning increased the data size.”
>
> ISTUDIO is the name of the assessment, i.e. the set of questions. A student whose data is included in the ISTUDIO assessment results responded to all six question prompts, and there were 1,935 distinct students who submitted responses.  If all responses were complete and usable, this would give 11,610 individual item responses, but due to students who did not complete all questions, and data cleaning, the final count was 7,258 items.
>
> “Question A: why did not the author show the results using rublic? To avoid the suspicion of arbitrarily changing the experimental setup, they should have first shown the results of the experiment using all input types and then the ablation results.”
>
>
> We can add the ablation condition where we included use of the rubrics, where the result was (81.21) with a confidence interval of (78.14, 82.74). As we stated in the paper, this does not improve performance; in fact, it degrades performance.
>
> “The full name of the proposed method (i.e., "Answer-state Recurrent Relational Network") only appeared in the Abstract. It also should be written in the main text of the paper.”
> “The font size in Figure 2 is too small to understand what the figure means. In addition, the positions of Figure 2 and its description are too far apart.
> ”
> Good point, so we would improve the figures.

---

### Official Review · Reviewer_ui5s · 2023-08-04

**Typos Grammar Style And Presentation Improvements:** The symbol (\tao) in formulation of S…
**Soundness:** 3

**Excitement:**

2: Mediocre: This paper makes marginal contributions (vs non-contemporaneous work), so I would rather not see it in the conference.

**Paper Topic And Main Contributions:**

The paper presents an automated method for evaluating constructed responses from the students. The specific assessment situation addressed in this work includes a prompt (context) against which two questions are asked. The first question seems to be an affirmative type while the second question asks for an explanation behind the answer in the first part. The data contains rubric and reference answers for three categories: Correct, Partially correct and incorrect. The authors advocated for feedback grouping. A recurrent relational network is used on a computation graph formulated considering the dependency between elements (context, question, reference answer etc.) present in the evaluation bundle. Several baselines have been used to compare the performance of the proposed model.

**Questions For The Authors:**

What is the state in ‘answer-state’?

It seems that the partially correct answers have been chosen to be the candidates for providing formative feedback. However, in a learning setting, incorrect answers demand equal scaffolding demand if not greater. Why is feedback for incorrect answers ignored?

On the formative feedback part, it is observed that AsRNN has good agreement with B on 2b and with A on 4b. But why is this observation an indicator of high accuracy of AsRNN? To illustrate more, A and B have poor agreement in 4b. Good agreement of RN with A does not ensure robustness of AsRNN. Consider a hypothetical situation where A’s annotation is incorrect and B’s annotation is correct. This will result in poor agreement between A and B.  Having good agreement between A and RN indicates that RN follows an erroneous annotator.


**Reasons To Accept:**

The work addresses an interesting and different aspect of automated assessment with respect to existing ASAG tasks and datasets. This task includes constructed response while the other ASAG task definitions mostly cover definition or concept completion type of questions.

The work also addresses the problem of selecting formative feedback. This aspect of grading has by and large been ignored in ASAG task definition of the earlier works.

The experiments include comparison with state-of-art LLMs on existing datasets in ASAG domain.


**Reasons To Reject:**

There is a lack of discussion regarding architectural decisions. For example, why is the Recurrent Relational Network chosen over other graph neural networks? The notion of ‘Answer-state’ in Answer-State RNN (AsRNN) has never been motivated or illustrated.

The contributions of the world have not been articulated clearly

The architectural novelty is incremental. Only adaptation over RNN seems to be using problem specific dependency rather than considering full connectivity.


**Reproducibility:**

3: Could reproduce the results with some difficulty. The settings of parameters are underspecified or subjectively determined; the training/evaluation data are not widely available.

**Reviewer Confidence:**

4: Quite sure. I tried to check the important points carefully. It's unlikely, though conceivable, that I missed something that should affect my ratings.

---

> ### Author Rebuttal · Authors · 2023-08-29
>
> Thanks for all your comments. We respond to your comments and questions below.
>
> “There is a lack of discussion regarding architectural decisions. For example, why is the Recurrent Relational Network chosen over other graph neural networks? The notion of ‘Answer-state’ in Answer-State RNN (AsRNN) has never been motivated or illustrated.”
>
> The choice of model architecture in this project is driven by a variety of factors, including the research problem, the nature of the data, and the specific requirements of the task.The Recurrent Relational Network (RRN) was chosen because it fits the the problem in four ways:
> 1. RRNs were developed for tasks that require a model to reason about relationships among multiple entities or concepts, where the learned relationships are represented as vectors with their own parameters. In contrast to attention networks, the number of parameters is kept low, so models are efficient. In our context, these relationships are among the variables C, Q, R, and A.
> 2. Recurrence over time helps RRNs continue to learn complex relationships. In complex scenarios that require temporal relationships to be modeled—as in our task where sequences like C->Q->R->A are evaluated and then loop back to reexamine C—the recurrent structure proves invaluable.
> 3. Palm et al. (2018) use a RRN to answer the questions about the relationships between objects. This reasoning capacity makes RRN well-suited to our problem, which demands reasoning about multiple relationships among the various types of text input.
> 4. Implementing an RRN from scratch is quite straightforward, because the conceptual structure and the implementation (from Palm et al.) are easy to understand, and it is also straightforward to develop many variants of an RRN from the same codebase. For example, the many ablation conditions required relatively few changes.
> Comparison with Other Graph Neural Networks (GNNs):
> We initially considered alternative GNN architectures such as Graph Convolutional Networks (Kipf et al. 2016, Wei et al. 2016) and Graph attention networks (Veličković et al. 2016). These methods employ neural network layers to aggregate information from a node's neighbors to update node representations directly. For example to update the representation (h_i) of node i, GCN uses h_i = W * (h_neighbor_1+ … … +  h_eighbor_n) + B * h_i, where W and B are learnable parameters. GNNs are well-suited for tasks in computer vision that need to learn one node’s representation, based on a high volume of neighboring nodes, and where the neighbors often represent different regions of the same depicted object (e.g, trunk of a tree). In contrast, our task pertains to a small number of total nodes that each represent a distinct kind of textual element (e.g., question prompt and a reference answer). Instead of merging information across similar neighbors through convolution, our goal is to learn the distinct way that a reference answer relates to a question prompt, and that a student answer relates to a reference answer. We believe RRNs are a good fit to our problem because they compute “messages” (i.e., the relation vectors) in one step, and then use the message to update the node representations (see equations (1) and (2) in our paper).
>
> Kipf, Thomas N., and Max Welling. "Semi-supervised classification with graph convolutional networks." arXiv preprint arXiv:1609.02907 (2016).
>
> Wei, Shih-En, et al. "Convolutional pose machines." Proceedings of the IEEE conference on Computer Vision and Pattern Recognition. 2016.
>
> Veličković, Petar, et al. "Graph attention networks." arXiv preprint arXiv:1710.10903 (2017).
> Palm, Rasmus, Ulrich Paquet, and Ole Winther. "Recurrent relational networks." Advances in neural information processing systems 31 (2018).
>
> Clarifying 'Answer-state':
> We chose the  term “Answer state” because our design choices aim for learning the best possible representation of the ‘answer’ node, through appropriate dependencies with the other nodes, and in particular with the supernode, which essentially represents an implicit global “answer state” at each iteration of the graph. The final output, as well as the application of contrastive learning techniques, is centered on the answer node's state.
>
> “The contributions of the would have not been articulated clearly”
>
> The final paragraph of our introduction states our contributions, but does not provide a succinct listing, e.g., as bullets. We can reframe this paragraph to state three key contributions: 1) The creation of a novel dataset with input jointly from statistics educators and NLP researchers, specifically designed for the automatic assessment of constructed-response (CR) questions that might have complex dependencies (e.g., on scenario contexts); 2) The development of an Answer-State Recurrent Relational Network (AsRRN) capable of evaluating CR questions of various types that is both effective and highly efficient at learning domain dependencies, such as that the meaning of a question prompt can depend in part on a scenario context; 3) The introduction of a new training paradigm that leverages contrastive learning to enhance performance and to provide greater interpretability of its decisions when student answers are partially correct, given that there are different ways for a student’s answer to miss the mark.
>
> “The architectural novelty is incremental. Only adaptation over RNN seems to be using problem specific dependency rather than considering full connectivity.”
>
> Our work is non-incremental in that we present two innovations that extend the inherent semantic flexibility and computational efficiency of relation networks, and a third innovation that leads to greater interpretability of model decisions.
>
> We apply RNNs to a semantically complex problem with five distinct categories of candidate text input (scenario context, question prompt, rubric reference answer, student answer), and experimentally demonstrate that an RNN that uses only four of these (all but the rubric; see response to reviewer 3), and where the RNN dependencies capture the domain relations among these text elements, outperforms a fully connected network both in accuracy, and in efficiency.  While we did not save this information we could rerun the comparison and report the lower accuracy & efficiency of the fully connected model.
>
> We introduce a super-node that is interconnected with all other nodes. This serves to stabilize the information flow throughout the graph network while also implicitly learning a global representation of the current answer state during each iteration over the entire graph.
>
> Because our training objective not only employs cross-entropy loss to optimize classification performance, but also incorporates a contrastive learning loss, our model can provide insight into decisions about student answers classified as partially correct, through representations that are relatively closer to one of several distinct ways to be partially correct. This latter element takes advantage of the availability of reference answers to enhance the quality of representation learning. Additionally, use of the Max() operator to push the learned representations towards one of a set of partially correct reference answers is an innovation that has not been used in previous versions of contrastive loss objectives.
>
>
> “It seems that the partially correct answers have been chosen to be the candidates for providing formative feedback. However, in a learning setting, incorrect answers demand equal scaffolding demand if not greater. Why is feedback for incorrect answers ignored?”
>
>
> Our collaborators in statistics education note that in the large introductory college-level statistics courses they are familiar with, questions used for formative assessment are intended to identify how well students have learned the concepts presented so far, and that written feedback is much more effective for students who provide partially correct answers, and at the same time much more time consuming for instructors.  Students whose answers are correct have learned the contact, and any written feedback would be limited to something like “Good job.” Students whose answers are incorrect without an indication of partial understanding similarly tend to receive uniform feedback, which is that they need to revisit the relevant material and to get extra help.  Students whose answers are partially correct are best served when the feedback points out the part they got correct, the part they got incorrect, and how to think about ways to avoid the same misunderstanding in the future.
>
> “On the formative feedback part, it is observed that AsRNN has good agreement with B on 2b and with A on 4b. But why is this observation an indicator of high accuracy of AsRNN? To illustrate more, A and B have poor agreement in 4b. Good agreement of RN with A does not ensure robustness of AsRNN. Consider a hypothetical situation where A’s annotation is incorrect and B’s annotation is correct. This will result in poor agreement between A and B. Having good agreement between A and RN indicates that RN follows an erroneous annotator.
> ”
>
>  The point of this analysis is to illustrate that accuracy is not the relevant criterion, because expert assessors do not always agree, as our data shows.  However, when expert assessors do agree, it is an opportunity to demonstrate that AsRRN can be considered as another expert whose judgment sometimes agrees well with humans who agree, and sometimes does not. Future work will explore ways to build upon AsRRN's ability to utilize partially correct reference answers.
>
> “The symbol (\tao) in formulation of S_j and S_j^\prime is not defined.
> ”
>
> The symbol τ (\tao) represents the hyperparameter for temperature in Contrastive Learning.

---

### Official Review · Reviewer_DeHo · 2023-08-11

**Soundness:** 3

**Excitement:**

3: Ambivalent: It has merits (e.g., it reports state-of-the-art results, the idea is nice), but there are key weaknesses (e.g., it describes incremental work), and it can significantly benefit from another round of revision. However, I won't object to accepting it if my co-reviewers champion it.

**Paper Topic And Main Contributions:**

This work addresses a CR type new to NLP but common in college STEM, consisting of multiple questions per context. Answer-state Recurrent Relational Network (AsRNN) is proposed, which exploits the graph structure to learn temporal dependencies and contrastive learning for better representation learning. A new dataset is proposed.

**Reasons To Accept:**

1. Overall the paper gives good and clear explanation and implementation of AsRNN. Well-structured and smoothly.
2. The design choices are well-justified with explanations as well as as ablation study.
3. A new solution to the new task for NLP and release a new dataset.

**Reasons To Reject:**

1. The novelty of the proposed method is limited.
2. The dataset size of this paper seems like small to ensure generalizability.
3. More analysis on the training/ inference time complexity are needed.


**Reproducibility:**

4: Could mostly reproduce the results, but there may be some variation because of sample variance or minor variations in their interpretation of the protocol or method.

**Reviewer Confidence:**

2: Willing to defend my evaluation, but it is fairly likely that I missed some details, didn't understand some central points, or can't be sure about the novelty of the work.

---

> ### Author Rebuttal · Authors · 2023-08-29
>
> Thanks for all your comments. We respond to your comments and questions below.
>
> “The novelty of the proposed method is limited.”
>
> Like many published works, AsRRN builds up and combines insights from prior approaches.  It's main novelty is greater interpretability of the models' decisions for an important application.  Three specific innovations we pointed to are:
>
> 1. We presented a problem that is novel to NLP, and important for post-secondary education.
>
> 2. The final structure of AsRRN, verified through ablation experiments, demonstrates how AsRRN incorporates the domain semantics as prior knowledge to boost performance, an insight that could benefit other kinds of NLP problems.
>
> 3. While in general, learned representations are uninterpretable, we introduce greater interpretability of the models' decisions on partially correct items, which is a particular challenge for STEM educators in formative assessment of written work.
>
>
> “The dataset size of this paper seems like small to ensure generalizability.”
>
>
> Domain-specific tasks can have unique constraints that make such large datasets impractical or impossible to collect, but where smaller datasets still have value. Collecting and annotating classroom data, for example, is intrinsically labor-intensive and constrained by various factors such as ethical considerations, quality of annotation, and the classroom environment. Our Col-STAT dataset, with 6,532 entries, is substantial, when compared to other notable datasets that have sizes ranging from several hundred to at most 20K: Mohler (2009): 630, Mohler (2011): 2,273, ASAP-SAS(2012): 18,000, Beetle (2013): 5,652, SciEntsBank (2013): 10,598, Arabic (2014): 610, McDonald (2017): 924, Turkish (2020): 984.
>
> Moreover, as highlighted by the results in Table 2, our model generalizes across different runs, even when different development sets are used. Finally, we believe the dataset's generalizability for STEM Assessment—should not be solely gauged by its size. As discussed in Section 3, the data were meticulously collected from actual classrooms and annotated by domain experts, which adds significant value in terms of accuracy and validity, meaning the prior evidence we cited shows that this assessment measures what it is intended to measure, i.e., statistical reasoning skills.
>
> Mohler (2009): Mohler, M., R. Mihalcea. Text-to-text semantic similarity for automatic short answer grading. In Proceedings of the 12th Conference of the European Chapter of the ACL (EACL 2009), pages 567–575. 2009.
> Mohler (2011): Mohler, M., R. Bunescu, R. Mihalcea. Learning to grade short answer questions using semantic similarity measures and dependency graph alignments. In Proceedings of the 49th annual meeting of the association for computational linguistics: Human language technologies, pages 752–762. 2011.
> ASAP-SAS(2012): Shermis, M. D. Contrasting state-of-the-art in the machine scoring of short-form constructed responses. Educational Assessment, 20(1):46–65, 2015.
> SciEntsBank (2013), Beetle (2013):Dzikovska, M. O., R. Nielsen, C. Brew. Towards effective tutorial feedback for explanation questions: A dataset and baselines. In Proceedings of the 2012 Conference of the North American Chapter of the Association for Computational Linguistics: Human Language Technologies, pages 200–210. 2012.
> Arabic (2014): Gomaa, W. H., A. A. Fahmy. Automatic scoring for answers to arabic test questions. Computer Speech & Language, 28(4):833–857, 201
> McDonald (2017): McDonald, J., R. Bird, A. Zouaq, et al. Short answers to deep questions: supporting teachers in large-class settings. Journal of Computer Assisted Learning, 33(4):306–319, 2017.
> Turkish (2020): Çınar, A., E. Ince, M. Gezer, et al. Machine learning algorithm for grading open-ended physics questions in Turkish. Education and Information Technologies, 25(5):3821–3844, 2020.
>
> “More analysis on the training/ inference time complexity are needed.”
>
> We trained AsRRN on an NVIDIA RTX A5000 GPU. On Col-STAT, the training fluctuated between 2 to 4 hours, depending on factors such as the ablation condition, and number of time steps in the relation network recurrence. The inference time was consistently under 3 minutes, emphasizing the model's efficiency. Training times on SciEntsBank ranged from 3 to 5 hours depending on the settings, showcasing that even with a dataset approximately twice as large, there is only a 50% to 25% increase in training time.

---

### Meta-Review · Area_Chair_LWqf · 2023-09-19

**Recommendation:** 1

**Metareview:**

The reviewers find the task interesting and the paper well-written. However, the reviewers raised some issues with the size of the dataset used; the lack of discussion about architectural decisions; and some found the STEM-specific terminology difficult to understand.

Overall, the reviewers find the soundness to be borderline, trending good, but are generally ambivalent about seeing the work published at EMNLP.

---

### Decision · Program_Chairs · 2023-10-07

**Decision:**

Accept-Findings

**Comment:**

The reviewers find the task interesting and the paper well-written. However, the reviewers raised some issues with the size of the dataset used; the lack of discussion about architectural decisions; and some found the STEM-specific terminology difficult to understand.

Overall, the reviewers find the soundness to be borderline, trending good, but are generally ambivalent about seeing the work published at EMNLP.